# Omicron-specific mRNA vaccination alone and as a heterologous booster against SARS-CoV-2

Zhenhao Fang[1,2,3,18], Lei Peng[1,2,3,18], Renata Filler[4,5], Kazushi Suzuki[1,2,3], Andrew McNamara[4,5], Qianqian Lin[1,2,3], Paul A. Renauer[1,2,3,6], Luojia Yang[1,2,3,6], Bridget Menasche[4,5,7], Angie Sanchez[1,2,3,8], Ping Ren[1,2,3], Qiancheng Xiong[9,10], Madison Strine[4,5,7], Paul Clark[1,2,3], Chenxiang Lin[9,10,11], Albert I. Ko[12,13], Nathan D. Grubaugh[12,14], Craig B. Wilen[4,5,7✉] & Sidi Chen[1,2,3,6,7,15,16,17✉]

The Omicron variant of SARS-CoV-2 recently swept the globe and showed high level of immune evasion. Here, we generate an Omicron-specific lipid nanoparticle (LNP) mRNA vaccine candidate, and test its activity in animals, both alone and as a heterologous booster to WT mRNA vaccine. Our Omicron-specific LNP-mRNA vaccine elicits strong antibody response in vaccination-naïve mice. Mice that received two-dose WT LNP-mRNA show a > 40-fold reduction in neutralization potency against Omicron than WT two weeks post boost, which further reduce to background level after 3 months. The WT or Omicron LNP-mRNA booster increases the waning antibody response of WT LNP-mRNA vaccinated mice against Omicron by 40 fold at two weeks post injection. Interestingly, the heterologous Omicron booster elicits neutralizing titers 10-20 fold higher than the homologous WT booster against Omicron variant, with comparable titers against Delta variant. All three types of vaccination, including Omicron alone, WT booster and Omicron booster, elicit broad binding antibody responses against SARS-CoV-2 WA-1, Beta, Delta variants and SARS-CoV. These data provide direct assessments of an Omicron-specific mRNA vaccination in vivo, both alone and as a heterologous booster to WT mRNA vaccine.

[1] Department of Genetics, Yale University School of Medicine, New Haven, CT, USA. [2] System Biology Institute, Yale University, West Haven, CT, USA.
[3] Center for Cancer Systems Biology, Yale University, West Haven, CT, USA. [4] Department of Laboratory Medicine, Yale University, New Haven, CT, USA.
[5] Department of Immunobiology, Yale University, New Haven, CT, USA. [6] Molecular Cell Biology, Genetics, and Development Program, Yale University, New Haven, CT, USA. [7] Immunobiology Program, Yale University, New Haven, CT, USA. [8] Yale College, New Haven, CT, USA. [9] Department of Cell Biology, Yale University, New Haven, CT, USA. [10] Nanobiology Institute, Yale University, New Haven, CT, USA. [11] Department of Biomedical Engineering, Yale University, New Haven, CT, USA. [12] Department of Epidemiology of Microbial Diseases, Yale School of Public Health, New Haven, CT, USA. [13] Department of Medicine, Section of Infectious Diseases, Yale University School of Medicine, New Haven, CT, USA. [14] Department of Ecology and Evolutionary Biology, Yale University, New Haven, CT, USA. [15] Comprehensive Cancer Center, Yale University School of Medicine, New Haven, CT, USA. [16] Stem Cell Center, Yale University School of Medicine, New Haven, CT, USA. [17] Center for Biomedical Data Science, Yale University School of Medicine, New Haven, CT, USA. [18] These authors contributed equally: Zhenhao Fang, Lei Peng. ✉email: craig.wilen@yale.edu; sidi.chen@yale.edu

Since its first identification in specimen collected in November 2021[1], the Omicron variant (lineage B.1.1.529) of SARS-CoV-2 has rapidly spread across the globe (Omicron cases tracker by Newsnodes; Tracking Omicron variant, GISAID)[2,3]. The Omicron variant was associated with increased risk of reinfection according to a population-level evidence in South Africa[4]. Two days after its initial report to World Health Organization (WHO), Omicron was designated as a variant of concern (VoC) by WHO on November 26 (Classification of Omicron (B.1.1.529): SARS-CoV-2 Variant of Concern. World Health Organization). Population-level data indicated that Omicron has become the dominant variant in South Africa in mid-November[5], only one week after the first traceable case. Similarly in January 2022, Omicron variant has dominated newly diagnosed cases in many states of the US (SARS-CoV-2 Omicron variant: statistics), Canada (Tracking variants of the novel coronavirus in Canada), and UK (Omicron cases tracker by Newsnodes). Omicron variant drove the fourth "wave" of Coronavirus Disease 2019 (COVID-19) in South Africa[6] and around the world[7]. Its case doubling time, every 3–4 days, is faster than previous waves[7], hinting its increased intrinsic transmissibility and/or immune evasion. A number of urgent questions on Omicron have quickly become central concerns, which include whether and when Omicron-specific vaccines or therapeutic antibodies will be effective against Omicron variant.

Mounting clinical and laboratory evidence have shown that most therapeutic and natural antibodies for COVID-19 failed to retain potency against Omicron[8–13]. The Omicron variant has 60 mutations compared to the ancestral variant's reference sequence (also referred to as prototypic virus/variant, reference, wild type (WT), Wuhan-Hu-1, or Wuhan-1, in lineage A; Tracking Omicron variant, GISAID). There are 50 nonsynonymous, 8 synonymous, and 2 non-coding mutations in Omicron, of which many are not observed in any other variants. There are a total of 32 mutations in the spike gene, which encodes the main antigen target of therapeutic antibodies and of many widely administered vaccines. This results in 30 amino acid changes, three small deletions, and one small insertion, of which 15 are within the receptor-binding domain (RBD; Implications of the emergence and spread of the SARS-CoV-2 B.1.1. 529 variant of concern (Omicron) for the EU/EEA). Due to its extensive number of mutations, this variant has high level of immune evasion, which drastically reduced the efficacy of existing antibodies and vaccines. Omicron spike mutations are concerning as they cluster on known neutralizing antibody epitopes[10] and some of them have well-characterized consequences such as immune evasion and higher infectivity. In fact, recent reports showed that the majority of the existing monoclonal antibodies developed against SARS-CoV-2 have dramatic reduction in effectiveness against the Omicron variant[8–13], leading to recall or exclusion of recommended use of certain therapeutic antibodies under emergency authorization (U.S. Pauses Distribution Of Monoclonal Antibody Treatments That Proved Ineffective Against Omicron).

The mRNA vaccines have achieved immense success in curbing the viral spread and reducing the risk of hospitalization and death of COVID-19[14,15]. However, a significant drop in mRNA vaccine's effectiveness against Omicron has been reported from clinical[5] and laboratory studies of samples of vaccinated individuals[16,17]. In light of the heavily altered antigen landscape of Omicron spike, assessing the efficacy of Omicron-specific mRNA vaccine is urgently needed. A number of critical questions regarding Omicron-specific mRNA vaccine need to be addressed. For examples: What is the immunogenicity of Omicron spike used in a vaccine form? Whether potent antibody immunity can be induced by an Omicron-specific mRNA vaccine, and how well does that neutralize the Omicron variant? If, and how, does the immune response induced by the Omicron-specific vaccine react to other variants, such as WA-1 (lineage A with a spike gene identical to WT or Wuhan-1) or Delta (lineage B.1.617.2)? Because a large share of world population received authorized Pfizer/BioNTech or Moderna mRNA vaccines that encode the reference (WT) spike antigen, it is important to know whether an Omicron mRNA vaccine can boost the waning immunity of existing vaccinated population. Clinical data showed that heterologous boosting with different types of COVID-19 vaccines elicited neutralizing titers similar to or greater than homologous boosting[18,19]. It is critical to compare the immunogenicity and efficacy of a heterologous Omicron booster with a homologous WT booster. Last but not least, as the antibody epitopes are closely related to their cross reactivity and susceptibility to variant mutations, it is crucial to know if and to what extent WT or Omicron mRNA vaccine can elicit plasma antibodies possessing broad antibody responses to SARS-CoV-2 variants and *Betacoronavirus* species.

To answer some of these questions, we directly generated an Omicron-specific lipid nanoparticle (LNP) mRNA vaccine candidate that encodes an engineered full-length Omicron spike with HexaPro mutations, and evaluated its effect alone, and compared its immunogenicity with WT LNP-mRNA as booster shots after SARS-CoV-2 WT mRNA vaccination in animal models.

## Results

**Design, generation, and physical characterization of an Omicron-specific LNP-mRNA vaccine candidate.** We designed an Omicron-specific LNP-mRNA vaccine candidate based on the full-length spike sequence of the Omicron variant (lineage B.1.1.529/BA.1) from two North America patients identified on November 23, 2021 (GISAID EpiCoV: EPI_ISL_6826713 and EPI_ISL_6826714). The spike coding sequence of Wuhan-Hu-1 (WT) and Omicron variant were flanked by 5′ UTR, 3′ UTR and 3′ PolyA tail (Fig. 1a). We introduced six Proline mutations (HexaPro) to the spike gene sequence, as they were reported to improve spike protein stability and prefusion state[20]. The furin cleave site (RRAR) in spike was replaced with GSAS stretch to keep integrity of S1 and S2 units. We then encapsulated the transcribed spike mRNA into lipid nanoparticles to produce WT and Omicron LNP-mRNAs, and characterized the quality and biophysical properties by downstream assays including dynamic light scattering, transmission electron microscope (TEM), and receptor-binding assay.

The dynamic light scattering and transmission electron microscope were applied to evaluate the size distribution and shape of Omicron LNP-mRNA, which showed a monodispersed sphere shape with an average radius of 52 nm and polydispersity index of 0.17 (Fig. 1c–e). To evaluate the effectiveness of LNP-mRNA mediated Omicron spike expression in cells as well as the receptor-binding ability of the designed Omicron HexaPro spike, Omicron LNP-mRNA was directly added to HEK293T cells 16 h before subjecting cells to flow cytometry. Evident surface expression of functional Omicron HexaPro spike capable of binding to human angiotensin-converting enzyme-2 (hACE2) was observed by staining cells with hACE2-Fc fusion protein and PE anti-Fc secondary antibody (Fig. 1f). These data showed that the Omicron spike sequence was successfully encoded into an mRNA, encapsulated into the LNP, can be introduced into mammalian cells efficiently without additional manipulation, and express functional spike protein that binds to hACE2.

**Specific binding and neutralizing antibody response elicited by Omicron LNP-mRNA against the Omicron variant.** After ensuring functional spike expression mediated by Omicron LNP-

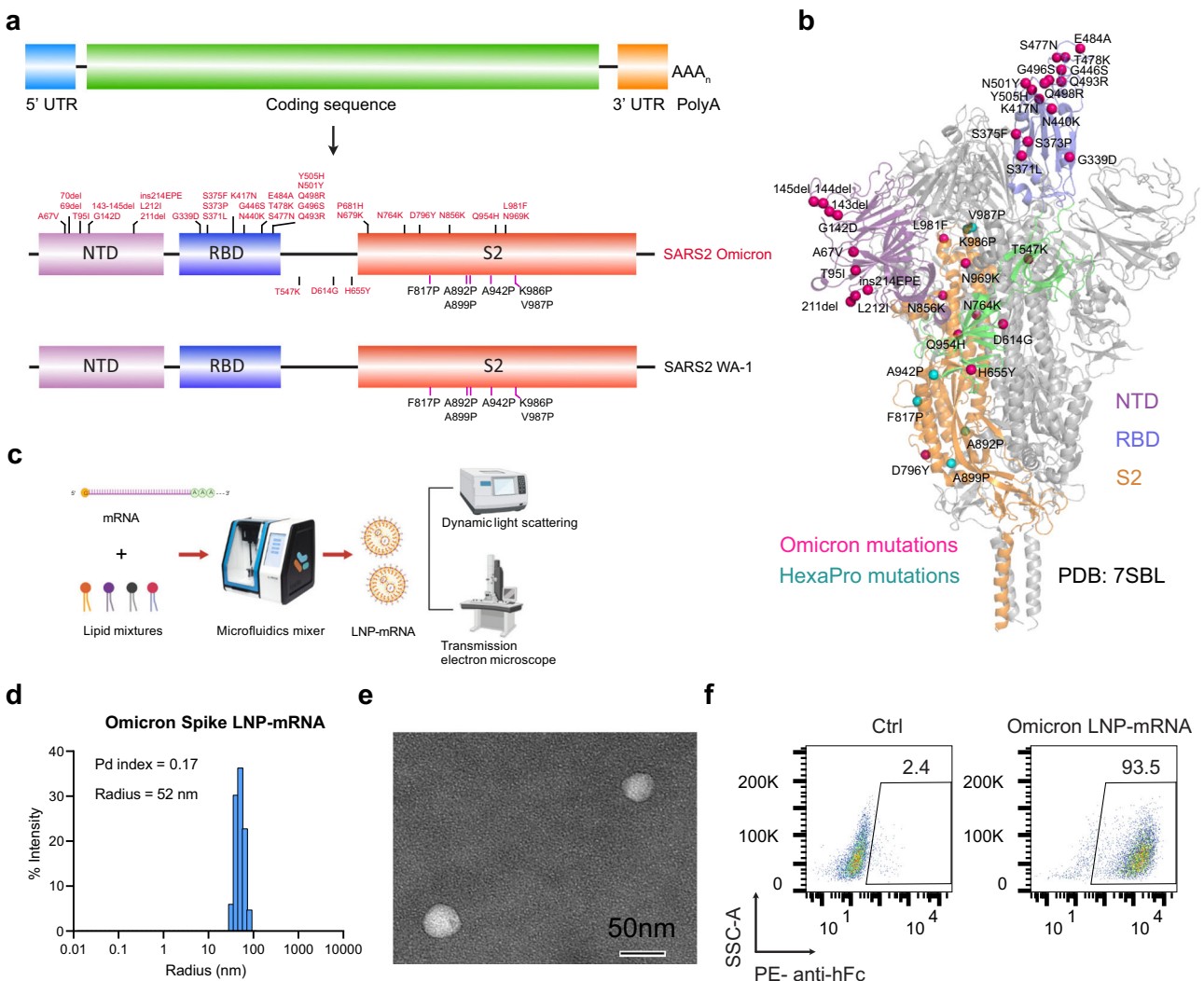

**Fig. 1 Design and biophysical characterization of Omicron-specific LNP-mRNA vaccine. a** Illustration of mRNA vaccine construct expressing SARS-CoV-2 WT and Omicron spike genes. The spike open reading frame were flanked by 5′ untranslated region (UTR), 3′ UTR, and polyA tail. The Omicron mutations (red) and HexaPro mutations (black) were numbered based on WA-1 spike residue number. **b** Distribution of Omicron spike mutations (magenta) were displayed in one protomer of spike trimer of which N-terminal domain (NTD), receptor-binding domain (RBD), hinge region and S2 were colored in purple, blue, green, and orange respectively (PDB: 7SBL). The HexaPro mutations in S2 were colored in cyan. **c** Schematics illustrating the formulation and biophysical characterization of lipid nanoparticle (LNP)-mRNA. Created with BioRender.com. **d** Dynamic light scattering derived histogram depicting the particle radius distribution of Omicron spike LNP-mRNA. **e** Omicron LNP-mRNA image collected on transmission electron microscope. **f** Human ACE2 receptor binding of LNP-mRNA encoding Omicron spike expressed in 293T cells as detected by human ACE2-Fc fusion protein and PE-anti-human Fc antibody on Flow cytometry.

mRNA, we proceeded to characterizing the immunogenicity of Omicron LNP-mRNA in vivo. In order to test rapid immune elicitation against Omicron variant, we performed the following vaccination and testing schedule. Two doses of 10 µg Omicron LNP-mRNA, as prime and boost two weeks apart were intramuscularly injected into ten C57BL/6Ncr (B6) mice (Fig. 2a and Supplementary Fig. 1a). Retro-orbital blood was collected prior to immunization on day 0, 13, and 21, i.e. 2 weeks post prime (one day before boost), and 1-week post boost. We then isolated plasma from blood, which was used in enzyme-linked immunosorbent assay (ELISA) and neutralization assay to quantify binding and neutralizing antibody titers. A significant increase in antibody titers against Omicron spike RBD was observed in ELISA and neutralization assays from plasma samples post prime and boost (Fig. 2b, c and Supplementary Fig. 1a, b). We performed neutralization with infectious virus (also commonly referred to as authentic virus or live virus) using a local SARS-

CoV-2 Omicron isolate in a biosafety level 3 (BSL3) setting (Methods section), and validated that the plasma samples from mice vaccinated with Omicron-specific LNP-mRNA showed potent neutralization activity against infectious Omicron virus, with significant prime/boost effect (Fig. 2d, e). These data showed that the Omicron LNP-mRNA induced strong and specific antibody responses in vaccinated mice.

**Waning immunity of WT LNP-mRNA immunized mice.** In light of the wide coverage of the ancestral WT-based LNP-mRNA vaccine (to model those widely administered in the current general population), we sought to test: (i) the effect of WT LNP-mRNA vaccination against Omicron variant, (ii) the decay of immunity induced by WT LNP-mRNA over time, and (iii) whether a homologous WT LNP-mRNA booster or a heterologous Omicron LNP-mRNA booster could enhance the waning immunity against Omicron variant, WA-1 and/or Delta variant,

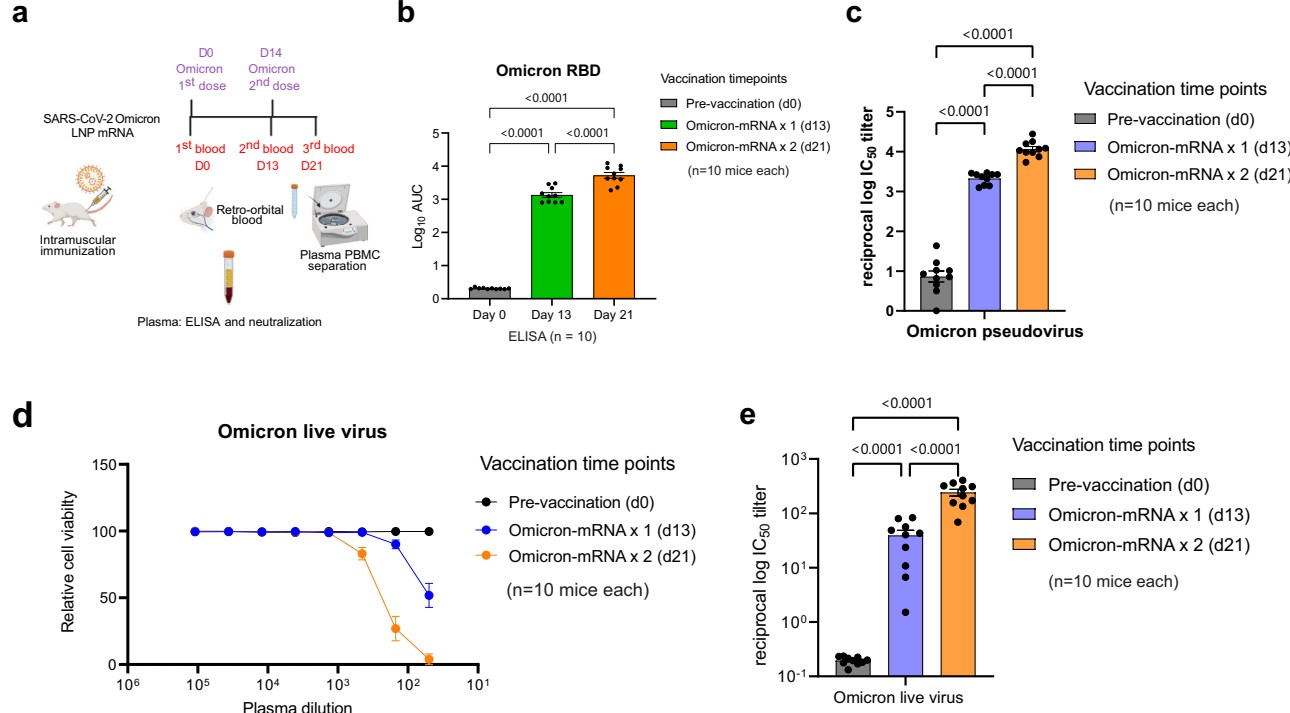

**Fig. 2 Omicron-specific LNP-mRNA vaccine-elicited neutralizing antibodies against SARS-CoV-2 Omicron variant. a** Immunization and sample collection schedule. Retro-orbital blood was collected prior Omicron LNP-mRNA vaccination on day 0, day 13, and day 21. Ten mice ($n = 10$) were intramuscularly injected with 10 μg Omicron LNP-mRNA on day 0 (prime, Omicron × 1) and day 14 (boost, Omicron × 2). The plasma and peripheral blood mononuclear cells (PBMCs) were separated from blood for downstream assays. The slight offset of the labels reflects the fact that each of the blood collections were performed prior to the vaccination injections. Data were collected from two independent experiments and each experiment has five mice. Created with BioRender.com. **b** Binding antibody titers of plasma from mice vaccinated with Omicron LNP-mRNA against Omicron spike RBD as quantified by area under curve of log10-transformed titration curve (Log10 AUC) in Supplementary Fig. 1. Each dot in bar graphs represents the value of one mouse ($n = 10$ mice). **c** Neutralization of Omicron pseudovirus by plasma from Omicron LNP-mRNA vaccinated mice. **d** Omicron live virus titration curves over serial dilution points of plasma from mice before and after immunization with Omicron LNP-mRNA at defined time points. Data of each sample were collected from three replicates ($n = 10$ mice). **e** Neutralization of Omicron infectious virus by plasma from Omicron LNP-mRNA vaccinated mice ($n = 10$ mice). Data on dot-bar plots are shown as mean ± s.e.m. with individual data points in plots. One-way ANOVA with Dunnett's multiple comparisons test was used to assess statistical significance. Statistical significance labels: *$p < 0.05$; **$p < 0.01$; ***$p < 0.001$; ****$p < 0.0001$.

and if there is a difference between homologous and heterologous boost. To gain initial answers to these questions in animal models, we sequentially vaccinated two cohorts of B6 mice with two doses of WT and one dose of WT or Omicron LNP-mRNA booster in two independent experiments (Batch 1 in Supplementary Fig. 2 and batch 2 in Supplementary Fig. 3). Over 100-day interval between 2nd dose of WT and WT/Omicron booster was ensured in order to observe the waning immunity in WT-vaccinated mice (the combined and individual datasets from the two independent experiments were presented in Fig. 3 and Supplementary Figs. 2–3 respectively). We collected blood samples of these animals in a rational time series, including day 35 (2 weeks post 2nd dose of WT LNP-mRNA), >3.5 months post 2nd doses of WT LNP-mRNA (day 127 in batch 1 or day 166 in batch 2, immediately before WT/Omicron booster), ~2 weeks post WT/ Omicron LNP-mRNA booster (day 140, one day before the second Omicron booster in batch 1 or day 180 in batch 2), and day 148 (1-week post two doses of Omicron LNP-mRNA vaccination in batch 1).

Plasma samples were isolated from blood samples and analyzed in ELISA and neutralization assays against SARS-CoV-2 Omicron, Delta or WA-1. Comparing to the titers against WA-1 and Delta RBD, the binding antibody titers against Omicron RBD elicited by WT mRNA-LNP were significantly weaker in samples from both day 35 and >3.5 months (Fig. 3b and Supplementary Figs. 4–5). The group average Omicron reactivity

is 15-fold (day 35) and 21-fold (>3.5 months) lower than that of WT RBD (fold change = ratio − 1), and 11-fold (day 35) and 14-fold (>3.5 months) lower than Delta (Supplementary Fig. 5). A steep (orders of magnitude) drop of antibody titers from mice immunized with WT LNP-mRNA was observed after three months (day 35 vs. >3.5 months) from all three RBD datasets. It is worth noting that the antibody titers >3.5 months post WT boost decreased to a level that is near-baseline (phosphate-buffered saline, PBS controls, Fig. 3), particularly for titers against Omicron RBD.

**Heterologous booster with Omicron LNP-mRNA as compared to homologous booster with WT LNP-mRNA in mice that previously received a two-dose WT LNP-mRNA vaccination.** A single dose booster shot, either a homologous booster with WT LNP-mRNA, or a heterologous booster with Omicron LNP-mRNA, drastically increased the antibody titers against Omicron RBD, by over 100-fold as compared to the sample right before booster shot (Fig. 3b), reaching a level comparable to the post-boost titer by Omicron LNP-mRNA alone (Fig. 2b). The mice that received the Omicron LNP-mRNA booster showed a trend of higher binding antibody titer against Omicron RBD than those administered with WT booster. Interestingly, the Omicron LNP-mRNA shot boosted not only titers against Omicron RBD, but also titers against Delta and WA-1 RBD, of which levels were comparable with those elicited by WT LNP-mRNA booster

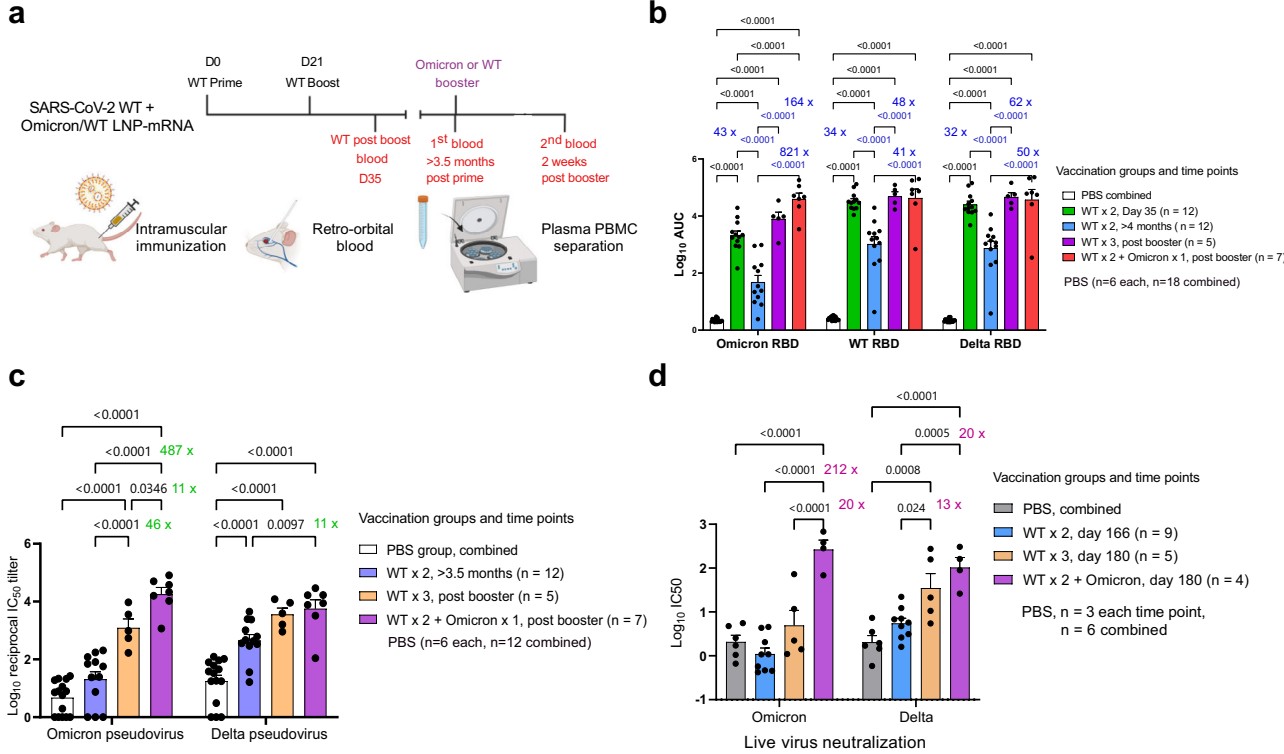

**Fig. 3 Heterologous booster with Omicron LNP-mRNA as compared to homologous booster with WT LNP-mRNA in mice that previously received a two-dose WT LNP-mRNA vaccination. a** Schematics showing the immunization and blood sampling schedule of mice administered with 1 μg WT LNP-mRNA prime (WT × 1) and boost (WT × 2) as well as 10 μg WT or Omicron-specific LNP-mRNA booster shots. The data was collected and combined from two independent experiments shown in Supplementary Fig. 2 and 3. Created with BioRender.com. **b** Bar graph comparing binding antibody titers of mice administered with PBS or WT and Omicron LNP-mRNA against Omicron, Delta, and WA-1 RBD (ELISA antigens). The antibody titers were quantified as $Log_{10}$ AUC based on titration curves in Supplementary Fig. S1a. PBS sub-groups ($n = 6$ each) collected from different matched time points showed no statistical differences between each other, and were combined as one group ($n = 18$). **c** Pseudovirus neutralizing antibody titers in the form of $log_{10}$-transformed reciprocal IC50 calculated from fitting the titration curve with a logistic regression model ($n = 12$ mice before booster, $n = 5$ in WT × 3, $n = 7$ in WT × 2 + Omicron). **d** Infectious virus neutralization titer comparisons between mice before and after vaccination with WT or Omicron boosters ($n = 9$ mice before booster, $n = 5$ in WT × 3, $n = 4$ in WT × 2 + Omicron). Titer ratios were indicated in each graph and fold change described in manuscript is calculated from (ratio − 1). Data on dot-bar plots are shown as mean ± s.e.m. with individual data points in plots. Two-way ANOVA with Tukey's multiple comparisons test was used to assess statistical significance. Statistical significance labels: *$p < 0.05$; **$p < 0.01$; ***$p < 0.001$; ****$p < 0.0001$. Non-significant comparisons are not shown, unless otherwise noted as n.s., not significant. Sample number is designated as n from biologically independent samples.

(Fig. 3b). For both WT and Omicron boosters, the extent of titer increase was more drastic in the Omicron RBD dataset than other RBD datasets, signifying the extra benefit of booster shots against Omicron variant (Fig. 3b). The antibody titers did not increase one week after a second booster of Omicron LNP-mRNA (Supplementary Fig. 2b).

Because pseudovirus neutralization is a relatively safer and widely-used assay that strongly correlates with infectious virus results and has been regarded as a standard proxy by the field[17,21–23], we set out to first use pseudovirus neutralization assay to measure the neutralizing antibody responses induced by Omicron LNP-mRNA booster in these animals. We first generated human immunodeficiency virus-1 (HIV-1) based Omicron pseudovirus system, which contains identical Omicron mutations in vaccine antigen, but lacks the HexaPro or furin site modifications. Interestingly, we found that under exactly the same virus production and assay conditions, the Omicron pseudovirus has higher infectivity than both WA-1 (8x increase) and Delta (4x) pseudoviruses (Supplementary Fig. 6a–c), which was also observed by another group[17], in concordance with the Omicron - hACE2 interactions from biophysical and structural studies[24,25], and correlated with higher transmissibility reported previously[17,26,27].

We then normalized the pseudoviruses by functional titers (number of infected cells/volume), and used this system to perform pseudovirus neutralization assays on all of plasma samples collected (Supplementary Fig. 6d, e). The neutralization results showed a consistent overall pattern as ELISA results, with a stronger contrast among titers against Omicron pseudovirus (Fig. 3c). On day 35 and >3.5 months post WT boost, the mice showed significantly lower neutralizing antibody titers against Omicron variant than titers against Delta variant or WA-1 (Supplementary Fig. 7a, b). For the samples two weeks post boost (day 35), the group average Omicron neutralization reactivity is 40-fold lower than that of WA-1 RBD, and 10-fold lower than Delta (Supplementary Fig. 7b). When comparing samples collected on day 35 and >3.5 months post WT boost, around two orders of magnitude (10 s–100 s of fold change) time-dependent titer reduction was unequivocally observed in all three pseudovirus neutralization data (Supplementary Fig. 2c). The Omicron-neutralization activity of WT-vaccinated mice >3.5 months post boost was as low as PBS background (Fig. 2c). These data suggested that there was waning antibody immunity in the standard two-dose WT-vaccinated animals, which lost neutralization ability against the Omicron variant pseudovirus.

A single booster shot of WT or Omicron LNP-mRNA vaccine enhanced the antibody titers against Omicron variant two weeks after the injection by >40-fold (Fig. 3c). The heterologous Omicron LNP-mRNA booster induced significantly higher neutralizing titer against Omicron pseudovirus than the homologous WT LNP-mRNA booster (Fig. 3c). The neutralizing titer after this surge by Omicron vaccine numerically surpassed the titer 2 weeks post WT vaccine boost (day 35, Supplementary Fig. 2c). Interestingly, the Omicron mRNA vaccine also rescued the antibody titers against Delta and WA-1 pseudoviruses, with two orders of magnitude increase in both ELISA titers and neutralization activity (Fig. 3b, c). The neutralization titers of Delta pseudovirus were found similar between WT and Omicron booster groups (Fig. 3c). A second booster shot two weeks after the first of Omicron mRNA vaccine yielded little increase in neutralization activity against Omicron, WA-1 or Delta variants at the time measured (day 148, 1 week after the second dose) (Supplementary Fig. 2c).

We went on to further evaluate the effects of WT and Omicron LNP-mRNA boosters in infectious virus neutralization assay, which closely correlated with pseudovirus neutralization results. The Omicron LNP-mRNA booster led to over 200-fold increase in neutralizing titers of infectious Omicron virus (Fig. 3d), while WT booster induced a moderate increase (10-fold) in titers against Omicron live virus (Fig. 3d). A significant boost of infectious Delta virus neutralizing titers was observed in mice receiving WT (12-fold) and Omicron (19-fold) LNP-mRNA boosters. A 20-fold difference in post-booster (day 180) neutralizing titers against infectious Omicron virus was observed between WT and Omicron booster groups (Fig. 3d). Together, these data suggest that while both WT LNP-mRNA and Omicron LNP-mRNA boosters can strengthen the waning immunity; however, the heterologous booster with Omicron-specific mRNA vaccination (WT × 2 + Omicron × 1) has an effect significantly stronger than the homologous booster (WT × 3) against the live virus of Omicron variant, with comparable activity against the Delta variant.

Overall, the ELISA titers, pseudovirus, and infectious virus neutralization activity were significantly correlated with each other across all groups and animals tested (Supplementary Fig. 9). These data suggested that a single dose of Omicron LNP-mRNA heterologous booster not only induced more potent anti-Omicron antibody response than WT booster, but also elicited broad activity against the WA-1 and Delta variant, in mouse models at the timepoints measured.

**Cross reactivity and epitope characterization of plasma antibodies from homologous Omicron mRNA, WT mRNA or heterologous WT + Omicron mRNA vaccination schemes.** In light of the broad activity elicited by heterologous vaccination of WT and Omicron LNP-mRNA, we ask if these vaccination schemes can induce antibody responses against other SARS-CoV-2 variants and other pathogenic *Betacoronavirus* species. We sought to answer these questions by characterizing and comparing the anti-coronavirus cross reactivity conferred by Omicron mRNA vaccination alone, WT mRNA vaccination alone (homologous booster), or their uses in combination (Omicron mRNA vaccination as a heterologous booster on top of WT mRNA vaccination). The cross reactivity was evaluated using six spike RBDs, including SARS-CoV-2 WA-1, Beta (lineage B.1.351) variant, Delta variant, Omicron variant, SARS-CoV spike RBD (SARS RBD) and MERS-CoV spike RBD (MERS RBD). Two doses of Omicron LNP-mRNA induced high titers of antibodies that cross reacted with all spike RBDs tested except for MERS RBD, which shared low sequence identity (<40%) to SARS or SARS-CoV-2 spikes (Fig. 4a). The antibody titer against SARS

RBD was significantly lower than those against SARS-CoV-2 WA-1 or variants (Supplementary Fig. 10a). Among the SARS-CoV-2 variants characterized, the antibody response to Delta variant by Omicron LNP-mRNA was slightly weaker than others. Both WT and Omicron boosters after WT LNP-mRNA prime and boost led to potent antibody response to SARS-CoV and SARS-CoV-2 Beta variant (Fig. 4b), while the response to MERS RBD was negligible and similar to PBS control. Within each ELISA antigen except for MERS RBD and Omicron RBD, the antibody response post WT or Omicron boosters (3 shots total) was numerically higher than that of plasma samples post a two-dose Omicron vaccine (Omicron × 2) (Supplementary Fig. 10c).

A number of studies have shown that antibodies whose epitopes overlap with hACE2-binding motif were largely escaped by RBD mutations in variants of concerns, while antibodies whose epitopes fall outside the hACE2-binding motif were rarer and often exhibit broad neutralizing activity to SARS-like *Betacoroanviruses* (*Sarbecoviruses*)[9,10,28,29]. Because of such correlation between antibody epitope and cross reactivity, we performed competition ELISA using hACE2 or antibodies with known epitopes as competing agents to evaluate the epitopes, population and affinity of plasma antibodies elicited by Omicron or WT LNP-mRNA. The epitopes of RBD can be categorized into several major classes based on cluster analysis of available neutralizing antibody-RBD complex structures[29–33]. We displayed representative antibodies in each major epitope class by aligning them with the recently solved Omicron RBD:hACE2 complex structure[24,25] (Fig. 4c). We then performed hACE2 and antibody competition ELISA using hACE2, Clone 13A, S309, and CR3022 as competing reagents to see if and to what extent group A-D[10] (class I–III[28], epitopes overlapped with hACE2) and group E–F (class IV, S309 and CR3022) antibodies were induced by these immunization schemes. Low-density Omicron RBD was coated in ELISA plate to ensure adequate competition between plasma antibodies and competing hACE2 or antibodies. In two independent experiments (hACE2 and antibody competition assays), the baseline titer of heterologous Omicron booster treated mice (WT × 2 + Omicron) in the absence of competing reagents was significantly higher than those of homologous WT booster treated mice (WT × 3), or mice receiving Omicron vaccination alone (Omicron × 2) (Fig. 4d). Addition of high concentration hACE2 (Methods) resulted in a significant reduction of plasma antibody titers in mice vaccinated with Omicron (Omicron × 2), WT (WT × 3) or WT + Omicron (WT × 2 + Omicron) LNP-mRNA (Fig. 4e). In the antibody competition assay, we used three antibodies with known RBD epitopes. Two of them (CR3022 and S309) are well-characterized representative antibodies from non-hACE2 competing classes. The Clone 13A is a humanized neutralizing antibody developed in our lab previously[34] and has an epitope that overlaps with the hACE2-binding motif. All three antibodies led to a significant decrease of plasma titers from Omicron vaccinated mice (Omicron × 2), while only CR3022 and S309 mediated a titer reduction in WT booster group (WT × 3) (Fig. 4f). The WT + Omicron heterologous vaccination group showed minimal titer changes to all three antibodies (Fig. 4f). These data suggested that a significant percentage of the pool of antibodies elicited by Omicron- or WT- vaccination shared binding epitopes with hACE2. In addition, antibody competition ELISA showed that both Omicron LNP-mRNA and WT LNP-mRNA vaccinated animals contained plasma antibodies targeting rare epitopes in class IV (or group E/F), which often exhibit broad activity against *Sarbecoviruses*.

## Discussion

The rapid spread of Omicron around the world, especially in countries with wide coverage of vaccines designed based on the

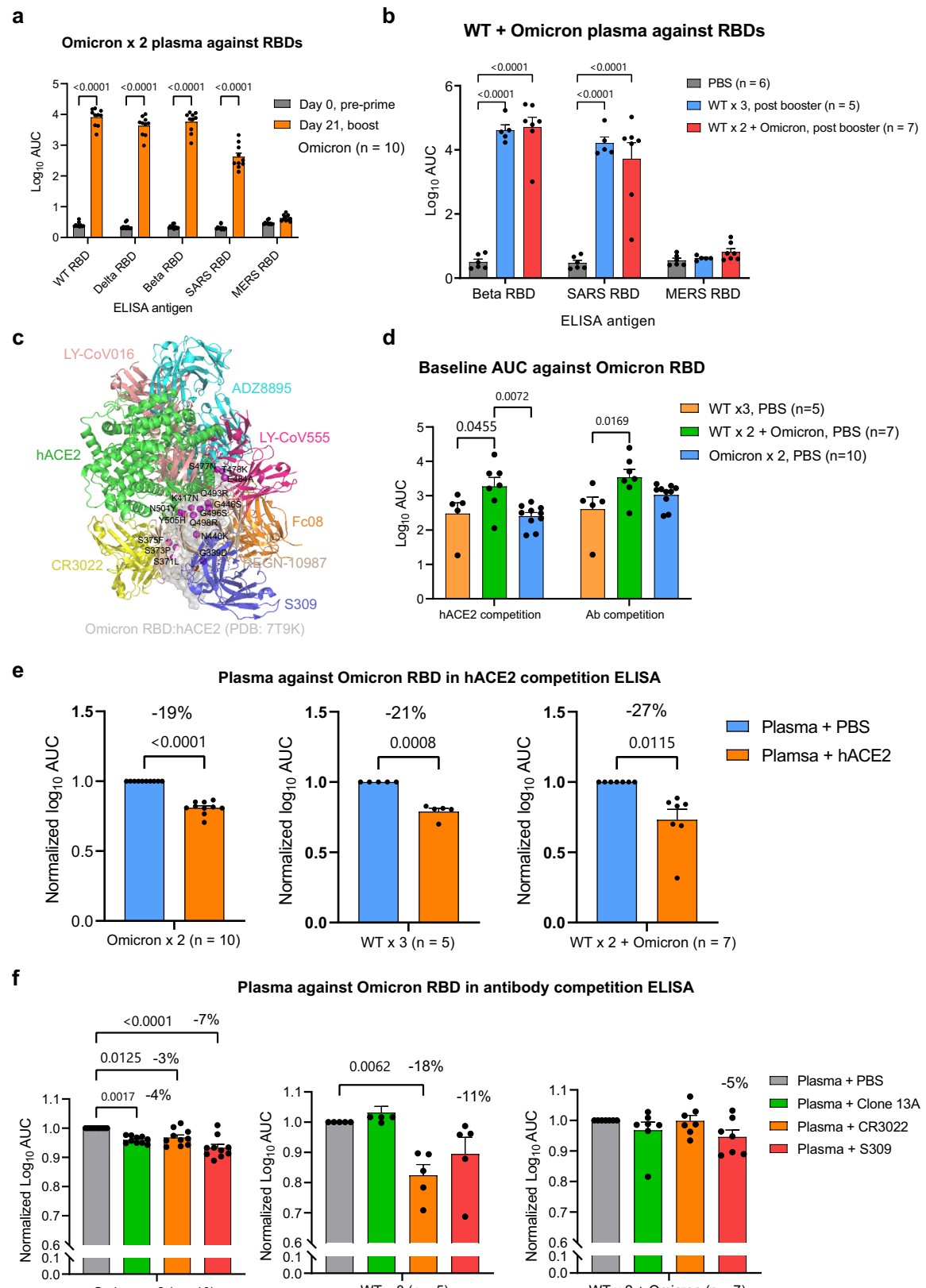

ancestral antigen (e.g. WT mRNA vaccine), is particularly concerning. The extensive mutations in the Omicron spike gene mark a dramatic alteration in its antigenicity[10]. Omicron has high transmissibility and high level of immune evasion from WT mRNA vaccine induced immunity, which was reported from various emerging literature[4,9–12]. Omicron's strong association

with reinfection[4] or breakthrough infection[5,7] and its heavily altered antigenicity prompted the idea of developing Omicron-specific mRNA vaccine.

To date (as of February 20, 2022), 4.35 billion people, i.e. 56% of the global population, received COVID-19 vaccination (Our World in Data; Coronavirus (COVID-19) Vaccinations). Almost

**Fig. 4 Cross reactivity and targeting sites characterization of plasma antibodies elicited by Omicron and WT LNP-mRNAs against SARS-CoV-2 VoCs and *Betacoronavirus* species. a** Cross reactivity of plasma antibody from mice immunized with Omicron LNP mRNA (prime and boost) to SARS-CoV-2 VoCs and pathogenic coronavirus species (n = 10 mice). **b** cross reactivity of plasma antibody from mice immunized with WT (WT × 3) or Omicron (WT × 2 + Omicron) boosters to SARS-CoV-2 beta variant and pathogenic coronavirus species (n = 6 mice in PBS, n = 5 in WT × 3, n = 7 in WT × 2 + Omicron). **c** Representative antibodies from major classes of RBD epitopes were shown by aligning spike RBDs in each of complex structures. The Omicron RBD surface was set to semi-transparent to visualize 15 RBD mutations and their relative positions to antibody epitopes. **d** baseline titers of plasma from mice of different vaccination status (WT × 3, WT × 2 + Omicron, Omicron × 2) were shown as $\log_{10}$ AUC determined in hACE2 and antibody competition ELISA. Each group sample number is denoted with n (n = 10 in Omicron × 2, n = 5 in WT × 3, n = 7 in WT × 2 + Omicron) in two independent assays (hACE2 and antibody competition ELISA). **e** Significant portion of plasma antibody from mice receiving Omicron (Omicron × 2, left panel) or WT + Omicron (WT × 3 middle, or WT × 2 + Omicron, right panel) LNP-mRNA competed with hACE2 for Omicron RBD binding in ELISA (n = 10 in Omicron × 2, n = 5 in WT × 3, n = 7 in WT × 2 + Omicron). **f** Plasma antibody from mice receiving Omicron (Omicron × 2, n = 10, left panel) or WT + Omicron (WT × 3, n = 5, middle or WT × 2 + Omicron, n = 7, right panel) LNP-mRNA showed various extent of binding reduction in the presence of blocking antibodies with known epitopes on RBD. The error bar and statistical information are identical with Fig. 3 and described in Method section.

all those vaccines were designed based on the antigen from the ancestral virus, including the two approved mRNA vaccine BNT162b2[14] and mRNA-1273[15]. Individuals receiving existing COVID-19 vaccines have waning immunity overtime[35–38]. Consistent with past reports, we observed a dramatic time-dependent decrease (around 40-fold) of antibody titers against Omicron, Delta variants, and WA-1 strains 3 months after the second dose of WT mRNA vaccine in mice. This observed waning immunity is particularly concerning in the scenario of rapid spreading of Omicron variant, which largely escapes the humoral immune response elicited by WT mRNA vaccines as evident in published studies[9–11,13] as well as in our current data. A recent report showed waning immunity in vaccinated individuals[17] and that a booster shot using the WT-based mRNA vaccine helps recover partial immunity. Our data showed that the neutralizing antibody titers after the boost with a WT-based vaccine were still lower against Omicron than against WA-1 and other variants, urging for development and testing of an Omicron-specific vaccine. Vaccinee receiving heterologous vaccination of WT and Omicron LNP-mRNA have been exposed to both antigens and may have robust antibody response against cognate strains and other VoCs. Thus, it is important to evaluate and compare the immunogenicity of Omicron-specific vaccine candidate with WT vaccine as booster shots on top of two doses of WT mRNA vaccine. In fact, very recently, both Pfizer and Moderna have started their clinical trials to evaluate the efficacy of Omicron-specific mRNA vaccine in either homologous or heterologous vaccination settings (Moderna starts trial for Omicron-specific booster shot; A Study to Evaluate the Immunogenicity and Safety of mRNA-1273.529 Vaccine for the COVID-19 Omicron Variant B.1.1.529)[39]. Moderna has released an updated Phase 2/3 clinical trial for their Omicron-specific mRNA vaccine (mRNA-1273.529) along with the WT vaccine mRNA-1273 against COVID-19 Omicron variant (NCT05249829). The scale and swiftness of initiating these clinical trials exemplify the clinical importance and urgent need of curbing the Omicron surge and evaluating the Omicron-specific mRNA vaccine.

In this study, we generated a HexaPro-version full-length Omicron spike LNP-mRNA vaccine candidate. In mouse models, we found that it can induce potent Omicron-specific and broad anti-Sarbecovirus antibody response. With this vaccine candidate, we compared its boosting effect with WT counterpart on animals that previously received two-dose WT mRNA vaccine. An observation is that a single dose of WT or Omicron boosters significantly strengthened the waning immunity against Omicron and Delta variants. A number of recent preprints generated and tested Omicron-specific vaccine candidates, which had different vaccine antigen designs, compositions, and showed varying results of antibody responses alone or as boosters[40–45]. Three of

them focused on evaluation of Omicron RBD mRNA vaccine alone in mice through neutralization assay and reported antibody response against Omicron but not other variants[40,43,44]. Two studies characterized the Omicron full-length spike mRNA stabilized by two proline mutations (S-2P) and compared their boosting efficacy with WT vaccine in mice[41] and macaque[42]. Preprints Gagne et al.[42] and Ying et al.[41] have shown that both WT and Omicron full-length spike mRNA boosters provided equivalent protection from Omicron challenge in non-human primates (NHPs) or mice. These results shared some commonalities, i.e. the effectiveness of an Omicron-specific vaccine; however, they diverged in the specific titers, as well as in the difference between WT- and Omicron-specific vaccines, potentially due to differences in vaccine antigen designs, compositions, modifications, experimental settings, animal models, or a combination of factors. Our study evaluated the potency of an Omicron-specific full-length spike mRNA vaccine with HexaPro mutations[20], which were shown to stabilize the spike in prefusion state. Through well-correlated data from ELISA, pseudovirus and infection virus neutralization assays, we showed that both WT and Omicron boosters significantly restored waning immunity against Omicron and Delta variants. Interestingly, without sacrificing potency against Delta, heterologous Omicron booster achieved significantly higher neutralizing titers against Omicron than homologous WT booster. This observation is in line with findings from heterologous booster vaccination of different COVID-19 vaccines in clinical trials[18,19]. The broad anti-coronavirus activity after homologous or heterologous boosting was likely associated with plasma antibodies in rarer epitope classes, as observed in competition ELISA.

It has been shown that the neutralizing antibody level is highly predictive of immune protection from SARS-CoV-2 infection and the initial neutralization level is associated with decay of vaccine efficacy over time[46]. Compared to WT booster, we found that Omicron booster group consistently showed 10–20-fold higher titers against Omicron variant in ELISA, pseudovirus and infectious virus neutralization assays. Within the WT-vaccinated group, the titer contrast against Omicron vs. Delta variants persisted over time. Omicron-booster group have been exposed to both WT and Omicron antigens and showed equally potent titers against Omicron and Delta. While our study is in animals, the antibody responses to vaccination are conserved between mouse and human, highlighted by the fact that mice are the main preclinical model used by vaccine developers[47,48].

The titer against Omicron by single dose Omicron LNP-mRNA was similar to that observed 2 weeks post boost of WT LNP-mRNA ($\log_{10}$ AUC or $\log_{10}$ IC50 around 3), although it is still unclear whether the potency of the Omicron mRNA vaccine is associated with the high number of Omicron mutations. As various extent of cross reactivity was observed among WT and/or

Omicron vaccinated animals, we sought to understand their cross-reactive immunity by characterizing vaccine-elicited antibody epitopes and population through competition ELISA. In the Omicron RBD competition ELISA, the baseline titer of Omicron LNP-mRNA booster group (WT × 2 + Omicron) was significantly higher than WT booster (WT × 3) or Omicron LNP-mRNA (Omicron × 2), which may explain its lower susceptibility to the block of competing antibodies. All three vaccination groups showed significant titer reduction in presence of hACE2, suggestive of abundant plasma antibody population sharing hACE2-binding epitopes, which are often associated with immune escape by variants mutations. The plasma from mice vaccinated with two doses of Omicron LNP-mRNA (Omicron × 2) or three doses of WT LNP-mRNA (WT × 3) exhibited comparable baseline titers and significant titer decrease when co-incubated with CR3022 or S309 blocking antibodies, indicating the existence of plasma antibody population sharing group E/F[10] or class IV[28] epitopes. Because of their similar baseline titers, the greater titer reduction in WT booster group may stem from larger population of group E/F antibodies, which was associated with higher cross-reactive response against SARS RBD (Fig. S10C). Albeit insignificant, the titer change of Omicron booster group (WT × 2 + Omicron) by S309 antibody was greatest among three competing antibodies, hinting a role of epitope IV antibodies in the cross immunity elicited by heterologous vaccination of WT and Omicron LNP-mRNA.

In summary, this study generated an Omicron-specific HexaPro spike LNP-mRNA vaccine candidate, studied its immunogenicity, and compared it with the WT counterpart in the context of previously WT-vaccinated animals. Our results showed that a single dose of either a homologous booster with WT LNP-mRNA or a heterologous booster with Omicron LNP-mRNA restored the waning antibody response, with over 200-fold titer increase by Omicron boosters. Interestingly, the heterologous Omicron LNP-mRNA booster elicited Omicron neutralizing titers higher than the homologous WT booster. The heterologous Omicron booster shot provided strong neutralizing antibody response against Omicron variant and comparable humoral antibody against WA-1 and Delta variants. All three types of vaccination, including Omicron mRNA alone, WT mRNA alone, and Omicron as a heterologous booster on top of WT mRNA, elicited broad antibody responses, including activities against SARS-CoV-2 VoCs, as well as other *Betacoronavirus* species such as SARS-CoV, but not MERS-CoV. Together, these data provided direct proof-of-concept assessments of Omicron-specific mRNA vaccination in vivo, both alone and as a heterologous booster to the existing widely-used mRNA vaccine form.

## Methods

**Molecular cloning**. The Omicron spike amino acid sequence was derived from two lineage BA.1 Omicron cases identified in Canada on November 23 2021 (GISAID EpiCoV, EPI_ISL_6826713, and EPI_ISL_6826714). Omicron spike cDNA were codon optimized, synthesized as gblocks (IDT) and cloned to mRNA vector with 5′, 3′ untranslated region (UTR) and poly A tail. The furin cleave site (RRAR) was replaced with a GSAS short stretch in the mRNA vector. HexaPro mutations were introduced in the WT sequence (Wuhan-Hu-1, which was used for the current clinical mRNA vaccines) and Omicron variant spike sequence of mRNA vector to improve expression and prefusion state[20]. The accessory plasmids for pseudovirus assay including pHIVNLGagPol and pCCNanoLuc2AEGFP were from Dr. Bieniasz' lab[49]. The C-terminal 19 amino acids were deleted in the SARS-CoV-2 spike sequence for the pseudovirus assay. A list of oligos has been provided in supplementary table 1.

**Cell Culture**. HEK293T (ATCC CRL-3216), HEK293FT (Thermo Fisher Cat. No. R70007), and 293T-hACE2 (gifted from Dr Bieniasz' lab) cell lines were maintained in Dulbecco's modified Eagle's medium (DMEM, Thermo fisher) supplemented with 10% Fetal bovine serum (Hyclone) and 1% penicillin-streptomycin (Gibco, final concentration penicillin 100 unit/ml, streptomycin 100 μg/ml), which is denoted as complete growth medium. Cells were split every 2 days at a split ratio

of 1:4 when the confluency reached over 80%. Vero-E6 cells were cultured in Dulbecco's Modified Eagle Medium (DMEM) with 5% heat-inactivated fetal bovine serum (FBS).

**In vitro mRNA transcription and vaccine formulation**. A Hiscribe™ T7 ARCA mRNA Kit (with tailing) (NEB, Cat # E2060S) was used to in vitro transcribe codon-optimized mRNA encoding HexaPro spikes of SARS-CoV-2 WT and Omicron variant with 50% replacement of uridine by N1-methyl-pseudouridine. The DNA template was linearized before mRNA transcription and contained 5′ UTR, 3′ UTR and 3′polyA tail as flanking sequence of spike open reading frame.

The purified mRNA was generated by following NEB manufacturer's instructions and kept frozen at −80 °C until further use. The lipid nanoparticles mRNA was assembled using the NanoAssemblr® Ignite™ instrument (Precision Nanosystems) according to manufacturers' guidance. In brief, lipid mixture composed of 46.3% ALC-0315 (MedChemExpress, HY-138170), 1.6% ALC-0159 (MedChemExpress, HY-138300), 9.4% DSPC (Avanti polar lipids, 850365 P), and 42.7% Cholesterol (Avanti polar lipids, 700100 P), was mixed with prepared mRNA in 25 mM sodium acetate at pH 5.2 on Ignite instrument at a molar ratio of 6:1 (LNP: mRNA)[47,50]. The LNP encapsulated mRNA (LNP-mRNA) was buffer exchanged to PBS using 100 kDa Amicon filter (Macrosep Centrifugal Devices 100 K, 89131-992). Sucrose was added as a cryoprotectant. The particle size of mRNA-LNP was determined by DLS device (DynaPro NanoStar, Wyatt, WDPN-06) and TEM described below. The encapsulation rate and mRNA concentration were quantified by Quant-iT™ RiboGreen™ RNA Assay (Thermo Fisher).

**Validation of LNP-mRNA mediated spike expression in vitro and receptor-binding capability of expressed Omicron HexaPro spikes**. On day 1, HEK293T cells were seeded at 50% confluence in 24-well plate and mixed with 2 μg Omicron LNP-mRNA. After 16 hours, the cells were collected for flow cytometry. The spike expression on cell surface were detected by staining cells with human ACE2-Fc chimera (Sino Biological, 10108-H02HG) in MACS buffer (D-PBS with 2 mM EDTA and 0.5% BSA) for 20 min on ice. Cells were washed twice after the primary stain and incubated with PE–anti-human Fc antibody (Biolegend, Cat. No. 410708, Clone No. M1310G05, 1:100 dilution) in MACS buffer for 20 min on ice. During secondary antibody staining, live/Dead aqua fixable stain (Invitrogen) was used to assess cell viability. Data was collected on BD FACSAria II Cell Sorter (BD) and analyzed using FlowJo software (version 10.7.2, FlowJo LLC).

**Negative-stain TEM**. Formvar/carbon-coated copper grid (Electron Microscopy Sciences, catalog number FCF400-Cu-50) was glow-discharged and covered with 6 μl of the sample for 1 min before blotting away the sample. The sample was double-stained with 6 μl of 2% (w/v) uranyl formate (Electron Microscopy Sciences, catalog number 22450) for 5 s (first stain) and 1 min (second stain), blotting away after each stain. Images were collected using a JEOL JEM-1400 Plus microscope with an acceleration voltage of 80 kV and a bottom-mount charge-coupled device camera (4k by 3k, Advanced Microscopy Technologies).

**Mouse vaccination**. All experiments in this vaccine immunogenicity study used 6–8-weeks-old female C57BL/6Ncr (B6) mice purchased from Charles River. The mice-housing condition was maintained at regular ambient room temperature (65–75 °F, or 18–23 °C), 40–60% humidity, and a 14 h:10 h day/night cycle. Each mice cage was individually ventilated with clean food, water, and bedding. Two sets of immunization experiments were performed: vaccination with Omicron LNP-mRNA, and sequential vaccination with WT LNP-mRNA, followed by WT or Omicron LNP mRNA booster. For the Omicron LNP-mRNA vaccination experiment, five mice were immunized with 10 μg Omicron LNP-mRNA on day 0 (prime) and day 14 (boost). Retro-orbital blood was collected prior to vaccine injection on day 0, day 13, and day 21. For WT and Omicron LNP-mRNA sequential vaccination experiment, 18 mice were administered with either 100 μl PBS (3 + 3 mice, two independent experiments) or two-dose 1 μg WT (on day 0 and day 21, 3 + 9 mice, two independent experiments) and 10 μg Omicron LNP-mRNA (over 3.5 months post prime). Retro-orbital blood was collected prior to booster shot or two weeks post booster and 2nd dose of WT LNP-mRNA.

**Institutional approval**. This study has received institutional regulatory approval. All recombinant DNA (rDNA) and biosafety work were performed under the guidelines of Yale Environment, Health and Safety (EHS) Committee with approved protocols (Chen 18–45, 20–18, and 20–26). All animal work was performed under the guidelines of Yale University Institutional Animal Care and Use Committee (IACUC) with approved protocols (Chen 2020-20358; Chen 2021-20068; Wilen 2021-20198).

**Isolation of plasma and PBMCs from blood**. At the defined time points, retro-orbital blood was collected from mice. The isolation of PBMCs and plasma was achieved via centrifugation using SepMate-15 and Lymphoprep gradient medium (StemCell Technologies). 200 μl blood was immediately diluted with 800 μl PBS with 2% FBS. The blood diluent was then added to SepMate-15 tubes with 6 ml Lymphoprep (StemCell Technologies). Centrifugation at 1200 × g for 20 min was

used to isolate RBCs, PBMCs and plasma. 250 µl diluted plasma was collected from the surface layer. The remaining solution at the top layer was poured to a new tube to isolate PBMCs, which were washed once with PBS + 2% FBS. The separated plasma was used in ELISA and neutralization assay.

**ELISA**. In all, 3 µg/ml of spike antigens were coated onto the 384-well ELISA plates (VWR, Cat # 82051-300) overnight at 4 degree. The antigen panel used in the ELISA includes RBDs of SARS RBD (AcroBiosystems, SPD-S52H6), MERS RBD (AcroBiosystems, SPD-M52H6), 2019-nCoV WA-1 (Sino Biological 40592-V08B), Delta variant B.1.617.2 (Sino Biological 40592-V08H90), Beta variant B.1.351 (Sino Biological 40592-V08H85) and Omicron variant B.1.1.529 (Sino Biological 40592-V08H121). Plates were washed with PBST (PBS plus 0.5% Tween 20) three times in the 50TS microplate washer (Fisher Scientific, NC0611021) and blocked with 0.5% BSA in PBST at room temperature for one hour. Plasma was fourfold serially diluted starting at a 1:500 dilution. Diluted plasma samples were added to the plates and incubated at room temperature for one hour, followed by washes with PBST five times. Anti-mouse secondary antibody (Fisher, Cat. No. A-10677) at 1:2500 dilution in blocking buffer was incubated at room temperature for one hour. Plates were washed five times and developed with tetramethylbenzidine substrate (Biolegend, 421101). The reaction was stopped with 1 M phosphoric acid after 20 min at room temperature, and OD at 450 nm was measured by multimode microplate reader (PerkinElmer EnVision 2105, Envision Manager v1.13.3009.1401). The binding response (OD450) was plotted against the dilution factor in log10 scale as the dilution-dependent response curve. The area under curve of the dilution-dependent response (Log10 AUC) was calculated to quantify the potency of the plasma antibody binding to spike antigens. The fold change of antibody titer was estimated using this equation: $ratio = 10 \wedge (AUC1 - AUC2)$.

**hACE2 and antibody competition ELISA**. The 384-well plate was coated with 0.6 µg/ml Omicron RBD at 4 degree overnight before washed with PBST (0.5% Tween-20) three times and blocked with 2% BSA in PBST for 1 h at room temperature. In hACE2 and antibody competition ELISA, 15 µg/ml hACE2 (Sino, 10108-H08H) or 10 µg/ml antibodies including Clone 13 A (Chen lab, in house), CR3022 (Abcam, Cat. No. Ab273073, Clone No. CR3022) and S309 (BioVision, Cat. No. A2266, Clone No. S309) were respectively added to the plate 1 hour prior to subsequent incubation with serially diluted plasma for another hour at room temperature. After coincubation of plasma and hACE2/antibodies, the plate was washed five times with PBST and incubated with anti-mouse secondary antibody with minimal cross reactivity with human IgG (Biolegend, Cat. No. 405306, Clone No. Poly4053, 1:2500 dilution). The plate was washed five times after 1-hour secondary antibody incubation and developed with tetramethylbenzidine substrate (Biolegend, 421101). The reaction was stopped with 1 M phosphoric acid after 20 min at room temperature, and OD at 450 nm was measured by multimode microplate reader (PerkinElmer EnVision 2105). The normalized AUC was calculated by normalizing the value with AUC determined in PBS group.

**Omicron, WA-1, and Delta pseudovirus production and characterization**. For the neutralization assay, HIV-1 based SARS-CoV-2 WA-1, B.1.617.2 (Delta) variant, and B.1.1.529 (Omicron) variant pseudotyped virions were packaged using a coronavirus spike plasmid, a reporter vector and a HIV-1 structural protein expression plasmid. The reporter vector, pCCNanoLuc2AEGFP, and plasmid expressing HIV-1 structural proteins (pHIVNLGagPol) were gifts from Dr Bieniasz's lab. The spike plasmid for SARS-CoV-2 WA-1 pseudovirus truncated C-terminal 19 amino acids (denoted as SARS-CoV-2-Δ19) and was from Dr Bieniasz' lab. Spike plasmids expressing C-terminally truncated SARS-CoV-2 B.1.617.2 variant S protein (Delta variant-Δ19) and SARS-CoV-2 B.1.1.529 variant S protein (Omicron variant-Δ19) were made based on the pSARS-CoV-2-Δ19. All pseudoviruses were produced under the same conditions. Briefly, 293FT cells were seeded in 150 mm plates, and transfected with 21 µg pHIVNLGagPol, 21 µg pCCNanoLuc2AEGFP, and 7.5 µg of corresponding plasmids, in the presence of 198 µl PEI (1 mg/ml, PEI MAX, Polyscience). At 48 h after transfection, the supernatant was filtered through a 0.45-µm filter, and frozen in −80 °C.

To characterize the titer of WA-1, Delta, and Omicron pseudoviruses packaged, $1 \times 10^4$ 293T-hACE2 cells were plated in each well of a 96-well plate. In the next day, different volumes of pseudovirus supplemented with culture medium to a total value of 100 µL were added into 96-well plates with 293T-hACE2. Plates were incubated at 37 °C for 24 h. Then cells were washed with MACS buffer once and the percent of GFP-positive cells were counted by Attune NxT Acoustic Focusing Cytometer (Thermo Fisher, Attune NxT Software v3.1). To normalize pseudovirus titer, $1 \times 10^4$ 293T-hACE2 cells were plated in each well of a 96-well plate. In the next day, 50 µL pseudovirus was mixed with 50 µL culture medium to 100 µL. The mixture was incubated for 1 hr in the 37 °C incubator, supplied with 5% CO2, and added into 96-well plates with 293T-hACE2. Plates were incubated at 37 °C for 24 hr. Then cells were washed with MACS buffer once and the percent of GFP-positive cells were counted by Attune NxT Acoustic Focusing Cytometer (Thermo Fisher). Delta pseudovirus and Omicron pseudovirus were diluted accordingly to match the functional titer of WA-1 pseudovirus for neutralization assay of plasma samples.

**Pseudovirus neutralization assay**. The SARS-CoV-2 pseudovirus assays were performed on 293T-hACE2 cells. One day before infection, $1 \times 10^4$ 293T-hACE2 cells were plated in each well of a 96-well plate. In the next day, plasma collected from mice were serially diluted by 5 fold with complete growth medium at a starting dilution of 1:100. 55 µL diluted plasma was mixed with the same volume of SARS-CoV-2 WA-1, Delta variant, or Omicron variant pseudovirus and was incubated for 1 hr in the 37 °C incubator, supplied with 5% CO2. 100 µL of mixtures were added into 96-well plates with 293T-hACE2. Plates were incubated at 37 °C for 24 h. Then cells were washed with MACS buffer once and the percent of GFP-positive cells were counted by Attune NxT Acoustic Focusing Cytometer (Thermo Fisher). The 50% inhibitory concentration (IC50) was calculated with a four-parameter logistic regression using GraphPad Prism (version 9.3.1, GraphPad Software Inc.). If the fitting value of IC50 was negative (i.e. negative titer), which suggested undetectable neutralization activity, the value was set to baseline (1, 0 in log scale).

**Omicron and Delta live virus production and characterization**. Full-length SARS-CoV-2 Omicron (BA.1) and Delta (B.1.617.2) isolates were a gift of Carolina Lucas and Akiko Iwasaki, and were isolated and sequenced[51]. Remnant nasopharyngeal swap samples selected for virus isolation were diluted in DMEM by 10 fold and then filtered through a 45-µm filter. Tenfold serial dilution of samples was made from 1:50 to 1:19,531,250. The diluted samples were subsequently co-incubated with TMPRSS2-Vero E6 in a 96-well plate and adsorbed for 1 h at 37 °C. Replacement medium was added after adsorption, and cells were incubated at 37 °C for up to 5 days. Supernatants from cells with cytopathic effect were collected, frozen, thawed and subjected to RT–qPCR.

To expand viral stocks, 107 Vero-E6 cells stably overexpressing ACE2 and TMPRSS2 were infected with SARS-CoV-2 at an MOI of ~0.01. The Omicron stock was collected 2 dpi, clarified by centrifugation ($450 \times g$ for 10 min), filtered through a 0.45-micron filter, and concentrated tenfold using Amicon Ultra-15 columns. To increase titer, the Delta stock was collected at 1 dpi, clarified, filtered, and used to infect $5 \times 10^7$ Vero-E6 cells overexpressing ACE2 and TMPRSS2. At 1 dpi, supernatant was harvested, clarified, filtered, and concentrated as above. Viral stocks were titered by plaque assay in Vero-E6 cells[52]. In brief, $7.5 \times 10^5$ and $4 \times 10^5$ Vero-E6 cells were seeded in each well of 6-well plates or 12-well plates. The media was replaced the next day with 100 µl of 10-fold serially diluted virus. Gentle rocking was applied to the plates incubated at 37 °C for 1 h. Subsequently, overlay media, DMEM with 2% FBS and 0.6% Avicel RC-581 was added to each well. At 2 dpi for SARS-CoV-2, plates were fixed with 10% formaldehyde for 30 min, stained with crystal violet solution (0.5% crystal violet in 20% ethanol) for 30 min, and then rinsed with deionized water to visualize plaques.

**Infectious virus neutralization assay**. The complements and other potential neutralizing agents were heat inactivated in mouse plasma prior to infectious virus neutralization assay. Mouse plasma samples were serially diluted, then incubated with SARS-CoV-2 Omicron live virus for 1 h at 37 °C. The Omicron live virus was isolated from nasopharyngeal specimens and sequenced as part of the Yale SARS-CoV-2 Genomic Surveillance Initiative's weekly surveillance Program in Connecticut[53]. After coincubation, plasma/virus mixture was added to Vero-E6 cells overexpressing ACE2/TMPRSS2. Cell viability was measured at 3dpi or 5dpi using CellTiter Glo.

**Statistics and reproducibility**. Standard statistical methods were applied to non-high-throughput experimental data. The statistical methods are described here, in figure legends and/or supplementary Excel tables. Data on dot-bar plots are shown as mean ± s.e.m. with individual data points in plots. Two-way ANOVA with Tukey's multiple comparisons test and one-way ANOVA with Dunnett's multiple comparisons test were used to assess statistical significance for grouped and non-grouped datasets respectively. Statistical significance labels: *$p < 0.05$; **$p < 0.01$; ***$p < 0.001$; ****$p < 0.0001$. Non-significant comparisons are not shown, unless otherwise noted as n.s., not significant. Sample number is designated as n from biologically independent samples. Prism (version 9.3.2, GraphPad Software Inc.) and RStudio (version 1.3.959, RStudio software company) were used for these analyses. Additional information can be found in the supplementary excel tables. Most of the data were collected from one independent experiment unless specifically stated otherwise in figure legends. Over 40 TEM micrographs were collected at various magnifications in one independent experiment and a representative micrograph was shown in Fig. 1.

**Schematic illustrations**. Schematic illustrations were created with Affinity Designer or BioRender.

**Replication, randomization, blinding, and reagent validations**. Biological or technical replicate samples were randomized where appropriate. In animal experiments, mice were randomized by littermates.

Experiments were not blinded.

Commercial antibodies were validated by the vendors, and re-validated in house as appropriate. Custom antibodies were validated by specific antibody - antigen

interaction assays, such as ELISA. Isotype controls were used for antibody validations.

Cell lines were authenticated by original vendors, and re-validated in lab as appropriate.

All cell lines tested negative for mycoplasma.

**Reporting summary**. Further information on research design is available in the Nature Research Reporting Summary linked to this article.

## Data availability

All data generated or analyzed during this study are included in this article and its supplementary information files. Specifically, source data and statistics are provided in a supplementary table excel file. No custom code was used in this study. Sequence of the Omicron variant (lineage B.1.1.529/BA.1) was derived from two North America patients in GISAID EpiCoV database with accession code of EPI_ISL_6826713 and EPI_ISL_6826714. Additional information related to this study are available from corresponding authors upon reasonable request.

## Code availability

The data collection and analysis of this study do not involve customized code. Codes that support this study are originated from publicly available software, as noted in the methods section.

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

## Acknowledgements

This work is supported by DoD PRMRP IIAR (W81XWH-21-1-0019) and discretionary funds to S.C.; and Ludwig Foundation, Mathers Foundation, and Burroughs Wellcome Fund to C.B.W. The TEM study is supported by NIH grant GM132114 to C.L.; Q.X. is supported by Singapore Agency for Science, Technology and Research Graduate Scholarship. We thank various members from our labs for discussions and support. We thank staffs from various Yale core facilities (Keck, YCGA, HPC, YARC, CBDS, and others) for technical support. We thank Drs. Tsemperouli, Karatekin, and others for providing equipment and related support. We thank Drs. Lucas and Iwasaki for sharing the Omicron virus isolate. We thank various support from Department of Genetics; Institutes of Systems Biology and Cancer Biology; Dean's Office of Yale School of Medicine and the Office of Vice Provost for Research.

## Author contributions

Z.F.: design of the study groups, constructs design and cloning, LNP prep, ELISA, data analysis, figure prep, and manuscript prep. L.P.: LNP prep, immunization, sample collection, neutralization, and data analysis. R.F. and A.M.: Live virus neutralization assay. K.S., Q.L., P.A.R., L.Y., A.S., P.R., and P.C.: assisting experiments, resources. Q.X.: TEM assistance. B.M. and M.S.: assisting BL3 experiments, resources. C.L.: TEM resource. A.K. and N.G.: Omicron isolate resource, manuscript prep. C.B.W.: BL3 neutralization assay design and supervision, manuscript prep, and funding. S.C.: conceptualization, overall design, manuscript prep, funding, and supervision.

## Competing interests

A patent application has been filed by Yale University related to the data described here (inventors: S.C., L.P., Z.F., and P.R.). Yale University has committed to rapidly executable nonexclusive royalty-free licenses to intellectual property rights for the purpose of making and distributing products to prevent, diagnose, and treat COVID-19 infection during the pandemic and for a short period thereafter. S.C. is a scientific Founder of EvolveImmune Tx and Cellinfinity Bio, unrelated to this study. The remaining authors declare no competing interests.
