## [Peer Review File · Nature Communications]

Omicron-specific mRNA vaccination alone and as a heterologous booster against SARS-CoV-2Editorial Note: This manuscript has been previously reviewed at another journal that is not operating a transparent peer review scheme. This document only contains reviewer comments and rebuttal letters for versions considered at Nature Communications.

Reviewers' Comments:

Reviewer #1:

Remarks to the Author:

The paper by Fang et al describes the generation and preclinical testing of a Omircon RNA based vaccines. I was not involved in the initial review of the manuscript, but it appears the authors have made a considerable effort to address prior reviewers comments, which has led to additional data that supports the overall conclusions of the paper. In my opinion, this paper is fit for publication.

Reviewer #2:

Remarks to the Author:

In this revised version of their manuscript, the authors went to a great length to satisfactorily answer all reviewer's comments.

I have two additional questions:

- what is the composition and/or source of the LNP used in these studies?
- Why did the authors apply different time intervals between prime and boost series in batch 1 vs batch ?

RESPONSE TO REVIEWERS' COMMENTS

Reviewer #1 (Remarks to the Author):

The paper by Fang et al describes the generation and preclinical testing of a Omircon RNA based vaccines. I was not involved in the initial review of the manuscript, but it appears the authors have made a considerable effort to address prior reviewers comments, which has led to additional data that supports the overall conclusions of the paper. In my opinion, this paper is fit for publication.

We thank reviewer 1 for this positive comment.

Reviewer #2 (Remarks to the Author):

In this revised version of their manuscript, the authors went to a great length to satisfactorily answer all reviewer's comments.

I have two additional questions:

- what is the composition and/or source of the LNP used in these studies?
- Why did the authors apply different time intervals between prime and boost series in batch 1 vs batch 2?

We thank reviewer 2 for the great suggestions, which have improved the quality of our study.

The LNP is composed of 46.3% ALC-0315 (MedChemExpress, HY-138170), 1.6% ALC-0159 (MedChemExpress, HY-138300), 9.4% DSPC (Avanti polar lipids, 850365P) and 42.7% Cholesterol (Avanti polar lipids, 700100P). We have added this information in manuscript (page 17).

We thank reviewer for pointing out this time interval difference in the two independent experiments, which is due to the two batch of existing mice already vaccinated with WT LNP-mRNAs receiving new boosters. This time interval difference would introduce background noise at the pre-boost level. However, it does not impede us drawing the main conclusion from the two independent experiments, e.g. both WT and Omicron specific boosters greatly improve the neutralizing titers of WT-immunized animals. In addition, this time interval variation is not uncommon in clinical studies or the real world setting, due to various reasons in individuals' booster schedule.